# The Ponto-Geniculo-Occipital (PGO) Waves in Dreaming: An Overview

**DOI:** 10.3390/brainsci13091350

**Published:** 2023-09-20

**Authors:** Jin-Xian Gao, Guizhong Yan, Xin-Xuan Li, Jun-Fan Xie, Karen Spruyt, Yu-Feng Shao, Yi-Ping Hou

**Affiliations:** 1Key Laboratory of Preclinical Study for New Drugs of Gansu Province, Departments of Neuroscience, Anatomy, Histology, and Embryology, School of Basic Medical Sciences, Lanzhou University, Lanzhou 730000, China; gaojx18@lzu.edu.cn (J.-X.G.); yangzh2021@lzu.edu.cn (G.Y.); xxli19@lzu.edu.cn (X.-X.L.); xiejf@lzu.edu.cn (J.-F.X.); 2NeuroDiderot-INSERM, Université de Paris, 75019 Paris, France; karen.spruyt@inserm.fr; 3Sleep Medicine Center of Gansu Provincial Hospital, Lanzhou 730000, China

**Keywords:** rapid eye movement sleep, ponto-geniculo-occipital waves, dreams, mechanisms underlying dreaming, non-rapid eye movement sleep

## Abstract

Rapid eye movement (REM) sleep is the main sleep correlate of dreaming. Ponto-geniculo-occipital (PGO) waves are a signature of REM sleep. They represent the physiological mechanism of REM sleep that specifically limits the processing of external information. PGO waves look just like a message sent from the pons to the lateral geniculate nucleus of the visual thalamus, the occipital cortex, and other areas of the brain. The dedicated visual pathway of PGO waves can be interpreted by the brain as visual information, leading to the visual hallucinosis of dreams. PGO waves are considered to be both a reflection of REM sleep brain activity and causal to dreams due to their stimulation of the cortex. In this review, we summarize the role of PGO waves in potential neural circuits of two major theories, i.e., (1) dreams are generated by the activation of neural activity in the brainstem; (2) PGO waves signaling to the cortex. In addition, the potential physiological functions during REM sleep dreams, such as memory consolidation, unlearning, and brain development and plasticity and mood regulation, are discussed. It is hoped that our review will support and encourage research into the phenomenon of human PGO waves and their possible functions in dreaming.

## 1. Introduction

Dreams are images and experiences that people have while they sleep. In ancient cultures, dreams were believed to contain messages from the gods or omens of the future. In the late 19th century, dreams became a subject of study for psychologists and psychoanalysts. Sigmund Freud’s *The Interpretation of Dreams* [1] initially developed the most prominent psychoanalytic theory of dreams. This model has been quite influential in sleep research and continues to have strong adherents to this day. For Freud, the dream is a highly meaningful mental product that is the result of specific mental processes under the conditions of sleep. Dreams that have “hidden meanings” and “repressed desires” are well established in popular folk psychology. Although Freud theorized that the purpose of dreaming is wish fulfillment, there is little experimental evidence to support this concept.

Modern sleep science has evolved significantly since the late 1950s, largely due to the seminal discoveries of rapid eye movement (REM) sleep [2,3,4,5]. Aserinsky and Kleitman [2] found that 74% of awakenings from REM sleep resulted in the recall of a dream, as compared to only 9% of awakenings from non-REM (NREM) sleep. The linking of REM sleep to dreaming ushered in a new era in the study of dreams [6]. During REM sleep, in addition to rapid eye movements (REMs), cortical electroencephalogram (EEG) desynchronization (or activation), loss of muscle tone, and autonomic fluctuations [2,3,4,5], many other physiological and behavioral features have also been found in humans and other mammals. These include high-amplitude spiky potentials of ponto-geniculo-occipital (PGO) waves, high-amplitude hippocampal EEG theta waves, penile erections, sporadic limb twitching, increases in brain/body temperature, and an elevated arousal threshold [7,8,9,10,11,12,13,14,15,16,17]. In these characteristic components of REM sleep, PGO waves are undoubtedly powerful internal sensory signals that convey a large amount of information to the visual cortex and seem to “compose the song sheet of dreams” [18]. Accordingly, a “dream state generator”, located mainly in the pontine reticular formation and producing PGO waves, has been postulated to be the cause of both REMs and the periodic intrusion of new content into hallucinatory dreams [19,20,21,22].

This review summarizes the PGO waves involved in the proposed neural mechanisms of dreaming and memory. The hypothesized physiological roles of PGO waves in performance are discussed.

## 2. The Neural Mechanisms Underlying Dreaming

### 2.1. PGO Waves

PGO wave activity was first discovered in cats in the 1950s [5,23,24]. Because these local field potentials originate in the pons (P) and propagate to the lateral geniculate nucleus (G) of the visual thalamus and the occipital cortex (O), they are called PGO waves [18,25,26]. PGO waves occur just before the onset of REM sleep and continue throughout its duration. They are characterized as biphasic, sharp field potentials lasting 60–120 ms, with an amplitude of 200–300 μV, occurring as singlets and clusters (Figure 1). These spikes during REM sleep period are in parallel to eye saccades and are observed not only in cats [5,25,27,28,29] but also in rats [30,31], mice [32], in non-human primates such as macaques [33,34], baboons [35], and in humans [7,36,37,38]. Cholinergic/glutamatergic neurons in the pontine brainstem have been shown to generate PGO waves by burst firing. PGO waves look just like a message sent from the pons to the lateral geniculate nucleus (LGN), occipital cortex, and other brain regions, including the temporal and prefrontal cortices and the amygdala. The dedicated visual pathway of PGO waves could be interpreted by the brain as visual information, thereby leading to the visual hallucinosis of dreams. These waves are not only associated with dream production but also limit the cortex’s ability to process external inputs [39,40]. Similar signals are known to be widespread in posterolateral cortical regions of the human brain. These PGO waves are the best candidates, so far, for the long-sought dream stimuli, and as such, they must somehow contribute to our thinking about consciousness [41].

In humans, many invasive and non-invasive studies provide insight into how PGO waves occur during REM sleep and how they contribute to dreaming. For example, in two invasive studies in Parkinson’s disease patients, deep brain stimulation electrodes were surgically implanted in the pedunculopontine nucleus of the pontine tegmentum [36] and the subthalamic nucleus (STN) [7]. The results showed that PGO-like waves were observed during REM sleep. Much like feline models, PGO wave singlets and clusters were recorded within STN during pre-REM and REM sleep [7]. A single-neuron study using depth-electrode EEG in patients with epilepsy found that potentials in the medial temporal lobe with a morphology similar to feline PGO waves were reliably observed time-locked to REMs, providing considerable support for the hypothesis that PGO waves propagate throughout the brain [42]. Several non-invasive technologies such as functional magnetic resonance imaging (fMRI) and positron emission tomography (PET) provide indirect evidence to support the existence of PGO waves in humans. An fMRI study combined with polysomnographic (PSG) recording [43] showed that the pontine tegmentum, ventroposterior thalamus, primary visual cortex, and putamen and limbic areas were activated during REM sleep in association with REMs. Neural evidence for the existence of human PGO waves and a link between REM and dreaming is provided by the REM-related activation of the primary visual cortex in the absence of visual input from the retina. The existence of PGO wave-like activity in humans was also supported by another study using PET combined with hemodynamic recordings of PSG activity [44]. This study found remarkable activity during REM in the ventroposterior thalamus and V1 of the occipital cortex, with additional activity in a number of limbic regions and the parahippocampal gyrus. This supports the hypothesis that PGO waves play an important role in the generation of visual content during REM sleep.

Overall, much of the detailed understanding of PGO waves and their mechanisms still depends on animal studies. However, direct translation to humans may require further investigation and validation. Using noninvasive methods such as fMRI and magnetoencephalogram, the future of PGO wave research undoubtedly lies in broad studies that combine behavioral, pharmacological, physiological, and cognitive experiments in human subjects [38].

### 2.2. Activation–Synthesis Hypothesis

Hobson and McCarley, following the pioneering work of Jouvet [45], proposed the activation–synthesis hypothesis in 1977 [19]. This hypothesis is based primarily on microelectrode recordings of PGO waves in cats, which were found to occur primarily during REM sleep. The dreaming process consists of activation and synthesis. In short, the neural activity in the pons activates the brain, especially the LGN and visual cortex, to generate information during REM sleep. Under these conditions, in the absence of external stimuli, internally generated inputs randomly activate sensorimotor information, and the passive synthesis of this information (perceptual, conceptual, and emotional) creates dreams (Figure 2). During REM sleep, these internal inputs generated by PGO waves continuously activate the forebrain via the midbrain reticular formation and are ready to process information. In this state, most motor outputs are blocked to prevent movement according to the dream content, but the oculomotor and vestibular systems are activated to induce REMs. Meanwhile, external sensory inputs are also blocked, increasing the relative impact of the internal inputs to the brain and thus enhancing the dream image (Figure 2). Ten years later, the activation–synthesis hypothesis was improved by Seligman and Yellen. They added the involvement of emotional appraisal to the activation of the primary visual cortex and secondary cognitive elaboration to explain the mechanism of dream generation [46].

Based on this hypothesis, Crick and Mitchison later proposed that the function of dreaming during REM sleep is to remove certain undesirable modes of interaction in networks of cells in the cerebral cortex so that the trace in the brain of the unconscious dream is weakened, rather than strengthened, by the dream [20].

### 2.3. Activation, Input, and Modulation (AIM) Model

The activation–synthesis hypothesis of dreaming was updated and extended by Hobson et al. [21,22], and transformed into the AIM state-space model of the brain–mind isomorphism, where a conscious state can be understood as a point in a three-dimensional state space. In the AIM model, the variables consist of activation (A), input (I), and modulation (M) (Figure 3). (A) The level of brain activation can be defined as the average firing frequency of brainstem neurons as reflected by high and low frequency levels in the EEG. The high levels of A in REM sleep are a correlate of the mind’s ability to access and manipulate significant amounts of stored information from the brain during dream synthesis. (I) Input source is a measure of how much of the sensory data being processed are from external or internal sources. During REM sleep, internally generated PGO waves replace blocked external sensory input, activating sensory and affective centers that then prime the cortex for dream construction by stimulating the visual cortex with PGO waves. This can be estimated from the frequency of REMs in REM sleep, which is thought to reflect brainstem PGO and motor generator activity. (M) The brainstem neuromodulators that they release exert a broad chemical influence on the brain. It is well known that REM sleep is cholinergic potentiated and monoaminergic suppressed (Figure 3A). In addition, the involvement of γ-aminobutyric acid (GABA) and glutamate has also been suggested to participate in this process. This model postulates that the transition to the cholinergic state in REM sleep, during which cholinergic PGO burst cells in the brainstem release acetylcholine, alters the mnemonic capacity of the brain–mind and reduces the reliability of cortical circuits, increasing the likelihood that bizarre temporal sequences and associations are accepted as reality during dreaming [47].

Within the three-dimensional space of the AIM model, the coordinates differ according to wake, NREM, and REM sleep states. In wakefulness, (A) is high, (I) is externally dependent, and (M) is dominated by monoaminergic systems. In NREM sleep, (A) and (M) are lower, and I is both external and internal. In REM sleep, (A) is high, (I) is internal, and (M) is predominantly from the REM sleep cholinergic system (Figure 3B). As suggested by the AIM model, there is little hallucinatory activity during NREM sleep, but recent research suggests that dreams still occur [48].

### 2.4. Neuronal Mechanisms of REM Sleep Regulation

REM sleep is generally thought to be mediated by a neural network located primarily in the brainstem. More recently, the concept of REM sleep regulation has evolved. Several hypothalamic and forebrain networks, including newly identified neuropeptides such as orexin and melanin-concentrating hormone (MCH), have been implicated in both the control and the final expression of this behavioral state [14,16,17,49,50,51,52,53,54]. Firstly, a reciprocal interaction between REM on and REM off states has been proposed to occur in the brainstem. In this model, the cholinergic laterodorsal and pedunculopontine tegmental neurons (LDT/PPT) are REM-on cells. The serotonergic dorsal raphe nucleus and noradrenergic locus coeruleus neurons are REM-off cells [22,55,56] (Figure 3A). However, this reciprocal interaction may not be sufficient to produce REM sleep, as suggested by experimental evidence over the past decade [16,53]. The glutamateric sublaterodorsal tegmental nucleus (SLD) and the GABAergic lateral paragigantocellular nucleus (LPGi) have been subsequently found to act as REM-on neurons, whereas GABAergic ventrolateral periaqueductal gray matter (vlPAG) and lateral pontine tegmentum (LPT) act as REM-off neurons [16,53]. Next, in the hypothalamus, orexinergic neurons are generally REM-off and are suppressed during REM sleep [57]. A deficiency in this process may be a cause of narcolepsy [16,17]. MCH neurons act as REM-on neurons and play an important role in REM sleep onset and maintenance [16]. In the forebrain, a transient increase in dopamine in the basolateral amygdala during NREM sleep terminates NREM sleep and initates REM sleep [58]. Thus, the mutual inhibitory interactions between REM-on and REM-off neurons switch the brain state between NREM and REM sleep [54]. Taken together, the REM sleep-regulating circuits are widely distributed throughout the brain stem (midbrain, pons, and medulla) and the hypothalamus, and they involve a number of neurotransmitters and neuropeptides. As a result, the REM sleep regulatory circuitry is a highly robust and complex system.

### 2.5. Other Recent Mechanisms of Dreaming

Solms [59] has hypothesized that dreaming is controlled by forebrain mechanisms. It is suggested that the cholinergic brain stem mechanisms that control the REM state are only able to produce the psychological phenomena of dreaming through the mediation of a second, presumably dopaminergic, forebrain mechanism. The dopaminergic forebrain circuits arise from the neurons in the ventral tegmental area (VTA) and terminate in the amygdala, anterior cingulate gyrus, and frontal cortex. This neural circuit of the mesocortical–mesolimbic dopamine system has been implicated in dream generation, and has been described as the “SEEKING” or “wanting” command system to subserve emotional drive and motivation [59]. In addition, a reward activation model (RAM) recently hypothesized that activation of the mesolimbic dopaminergic system during sleep, particularly REM sleep, contributes to memory processes, REM sleep regulation, dream generation, and motivation [13,60]. Thus, the mesolimbic dopaminergic system seems to play an important role in dreaming, both in the dopaminergic forebrain mechanism and in the RAM [13,59,60,61].

The amygdala, a limbic structure associated with emotions, memory, and dreams, receives the dopaminergic projection. It serves as a node to integrate the regulation of REM sleep and causes the intermittent appearance of REM sleep-related dreams and REM sleep behavior disorder (RBD) [58,61,62]. The amygdala is very active in REM sleep, especially in humans [63,64]. It can influence the frequency of PGO waves during REM sleep. This suggests that it also plays a key role in setting the “emotional tone” for PGO activity [65]. A very recent study by Hasegawa et al. showed that a transient increase in dopaminergic input from the VTA to the basolateral amygdala during NREM sleep triggers NREM-to-REM transitions [58]. Next, a recent neuro-physio-pharmacological study showed that both cholinergic PPT neurons and dopaminergic substantia nigra (SN) neurons project to the amygdala and modulate REM sleep, in a conformal manner, indicating that REM sleep-related dreaming may be due to the phasic activation of amygdala neurons by phasic REM-onset neurons in the PPT and SN [62]. However, whether the dopamine system is necessary or sufficient for transitioning to REM sleep and for dreaming remains to be determined.

## 3. The Physiological Functions of Dreaming: The Involvement of PGO Waves during REM Sleep

### 3.1. Memory Consolidation

Oneiric production is a form of mental sleep activity that appears to be closely related to memory processes and cognitive elaboration [48,66,67]. Converging evidence suggests that dreaming is influenced by memory consolidation during sleep [68]. A number of studies have reported that PGO waves during REM sleep in rodents have been repeatedly associated with memory consolidation [69]. For example, PGO waves increased during REM sleep following learning tasks [70,71,72,73,74]. Artificially enhancing PGO waves by injecting carbachol prevented avoidance memory deficits during a period of REM sleep deprivation [75], while suppressing PGO wave generation in rats impaired avoidance memory retention during sleep [76]. It appears that the density of PGO wave activity is directly related to memory processes. A number of studies have reported an increase in PGO wave density following fear memory training in rats, which predicted overnight memory consolidation [70,71,72,73]. The success of fear extinction was recently shown to be predicted by PGO wave density during REM sleep [74]. In addition, increased activity of brain-derived neurotrophic factors and plasticity-related immediate early genes in the dorsal hippocampus was associated with PGO wave density after training. The selective elimination of PGO wave-generating cells prevents these increases, whereas the enhancement of PGO waves through cholinergic activation of these cells enhances the increases [72,73]. 

In general, PGO waves in the transition from NREM to REM sleep are considered to be the physiological signals that initiate and maintain REM sleep, constituting a state with characteristics distinct from both the preceding NREM sleep and the following REM sleep [77,78]. Pontine caudolateral parabrachial neuronal discharge has been found to contribute to the shift toward the two PGO-related states, the transition from NREM to REM sleep and REM sleep [34,77,79]. Of note, a recent study in macaque monkeys showed that PGO waves during the transition from NREM to REM sleep co-occurred with hippocampal sharp-wave ripples associated with memory consolidation [34]. In addition, the frequency of PGO waves during the transition from NREM to REM sleep is significantly lower than during REM sleep, which may be related to the fact that the duration of dreams during NREM sleep is usually shorter than that of dreams during REM sleep [80].

Furthermore, PGO waves are usually highly correlated with REMs [26,38,79,81]. The caudoventral pontine tegmentum has been shown to be responsible for the simultaneous generation of the PGO waves and correlated REMs during REM sleep [28]. REMs have experimentally revealed gaze shifts in the virtual world of REM sleep in mice, providing an external readout of an internal cognitive process occurring during REM sleep and manifesting the memory of a brief episode in life that is critical for survival [82,83]. In addition, PGO waves and REMs tend to be phase-locked to hippocampal and neocortical theta waves [84,85,86]. The PGO wave clusters during REM sleep have been shown to coincide with hippocampal theta oscillations, which are thought to cause a permanent enduring increase in synaptic strength, allowing for the consolidation of new memories while maintaining existing ones [34]. Additionally, Boyce et al. [87] optogenetically silenced medial septal GABAergic cells that drive hippocampal theta activity. This resulted in a specific attenuation of the memory-associated theta rhythm during REM sleep without disrupting sleep. They found that the selective silencing of these GABAergic neurons during REM sleep impaired the subsequent recognition of novel object locations and fear-conditioned contextual memory [87]. This suggests that theta activity during REM sleep plays a critical role in memory consolidation. Therefore, PGO wave clusters coupled with REMs and theta waves during REM sleep are involved in learning and memory functions and may underlie the realistic and vivid experience of dreams.

Although evidence from a number of studies shows that PGO waves are closely related to learning, cognition, dreaming, visual hallucination, sensorimotor integration, and synaptic plasticity in the brain areas through which they pass [38], their functions are still not fully understood.

### 3.2. Unlearning

One of the hypothesized functions of REM sleep is a process of “unlearning” [20,88]. These authors proposed that the function of dream sleep is the removal of certain unwanted memories from the cerebral cortex. During REM sleep, the unconscious dream traces act to weaken rather than strengthen memory. It is noteworthy that “we dream to forget” [20] is not the same as normal forgetting. Dreams are not simply forgotten; they are actively unlearned [80]. The unlearning mechanism modifies the cerebral cortex by changing the strength of individual synapses. Because an increase in synaptic strength is necessary to consolidate memory, unlearning weakens synaptic strength. Crick and Mitchison [20] postulated that PGO waves are produced by this dream generator in the brainstem as a random internal activation. In order to eliminate many of the unwanted memory traces, the random internal activation must occur repeatedly because it is less effective if it occurs only once. Thus, the PGO waves that occur repeatedly during REM sleep play a key role in this process. Furthermore, PGO waves may determine which memories are retained and which are erased. It has also been suggested that if this unlearning process does not work, people experience hallucinations, delusions, and obsessions, leading to a state similar to schizophrenia [20].

The unlearning theory is supported by a growing number of studies. For example, during REM sleep, the postsynaptic dendritic spines of layer V pyramidal cells in the mouse motor cortex are eliminated during development and motor learning. On the other hand, critical spines are strengthened and maintained [89]. The hypothalamic MCH neurons are known to be involved in the control of REM sleep and mood [90]. An anatomical and functional study in cats found that MCH neurons project to cholinergic pontine neurons [91]. When MCH was microinjected into the nucleus pontis oralis, there was a significant decrease in latency to REM sleep and a significant increase in the amount of REM sleep, accompanied by increased PGO wave activity and its duration. This suggests that the MCH system is involved in the regulation of REM sleep by modulating neuronal activity in cholinergic pontine neurons [91]. However, it has recently been shown that MCH neurons are also involved in the unlearning mechanism of REM sleep [92]. The activation or inhibition of MCH neurons by optogenetics and chemogenetics impaired or improved hippocampal-dependent memory, respectively. The activation of MCH nerve terminals in vitro reduced the firing of hippocampal pyramidal neurons by increasing inhibitory inputs. These results strongly suggest that the activation of MCH projections to the hippocampus during REM sleep actively contributes to forgetting [92]. The activity of hippocampal neurons during REM sleep has been suggested to play a key role in unlearning [93]. REM sleep serves to maintain or strengthen memories until they are transferred out of the hippocampus, whereupon they should be erased from this space-restricted short-term memory factory so that these synapses can be used to encode new associative memories [93]. PGO waves that are phase-locked to the theta oscillation of the hippocampus during REM sleep may be involved in this process [94]. However, the question of whether REM sleep plays a more critical role in the processing of emotional and procedural memories than other types of memories remains unanswered. One attractive hypothesis is that REM sleep is responsible for erasing negative emotional memories, and that this function is dysfunctional in depressed patients [95].

### 3.3. Brain Development and Plasticity 

REM sleep is known to be particularly abundant during early development. At birth, half or more of our sleep time is occupied by REM sleep, compared to <20% of sleep time in adults [96]. PGO waves during REM sleep in development might be an important central nervous system (CNS) stimulator during a period when wakefulness is limited in time and scope and stimulation opportunities are few [94,97,98]. The ascending impulses emanating from the brainstem during REM sleep may be required to promote neuronal differentiation, maturation, and myelination in higher brain centers [17]; that is, REM sleep deprivation during the early life of animals has been used to understand some functional mechanisms of PGO waves in brain development. Selective REM sleep deprivation further eliminates the endogenous stimulation of PGO waves [17]. For example, when REM sleep is dramatically reduced in postnatal rats over the course of 2 weeks, these REM sleep-deprived rats have a reduced brain size, hyperactivity, anxiety, attention, and learning difficulties in adulthood [99,100]. In addition, selective REM sleep deprivation for 1 week during the critical postnatal developmental period in kittens significantly reduces the size of noradrenergic neurons in the locus coeruleus [101] and the number of parvalbumin immunoreactive neurons in the LGN [102]. REM sleep deprivation in monocularly occluded kittens reduces neuronal size in the monocular segment of the LGN [103] and ocular dominance plasticity by inactivating a kinase critical for this plasticity [104]. The elimination of phasic PGO waves in the LGN during REM sleep enhances plasticity effects on cell size in the LGN [105]. This suggests that REM sleep neuronal activities may be necessary for normal LGN development. The development of PGO waves is further thought to contribute to the maturation of the thalamocortical pathway in early life [98,106]. Thus, the abundance of REM sleep in early life and its subsequent decline to lower levels in adulthood strongly suggests that REM sleep is an integral part of the activity-dependent processes that enable normal physiological and structural brain development [100,102,106,107,108].

Several lines of evidence have shown that PGO waves in REM sleep are associated with the regulation of neural plasticity [109,110,111]. Brain plasticity allows for the preservation of the ability to change, adapt, and learn in response to different environmental experiences and new demands. These processes occur with sleep cycles throughout life and begin in response to REM sleep in late fetal and early neonatal life [112]. PGO wave-associated cells are good candidates for generating or modulating plasticity in various brain structures [71], as they discharge high-frequency spike bursts during pre-REM and REM sleep [34]. Indeed, the activation of PGO wave-generating cells by cholinergic agonism induces changes to the electrical properties of PGO wave activity [113], accompanied by prominent behavioral effects [74,114]. Furthermore, the activation of the PGO wave generator and the occurrence of PGO waves themselves are tightly correlated with increased levels of cAMP response element binding (CREB) proteins, brain-derived nerve growth factor (BDNF), and activity-regulated cytoskeletal protein (ARC) in structures including the hippocampus and amygdala [72]. An investigation of the functions and underlying mechanisms of REM sleep during postnatal development in mice revealed that REM sleep plays a fundamental role in establishing the stable connectivity of synaptic circuits during development. Almost all newly formed synapses cannot be maintained without REM sleep [89]. In addition, REM sleep plays an important role in a developmentally regulated form of in situ long-term potentiation (LPT) that coincides with the visual critical period [115]. During the visual critical period, PGO waves, especially the migratory waves of neural activity that can resemble evoked visual activity in sensory cortex [40], triggers a series of intracellular cascades and the synthesis of several plasticity-related proteins that promote cortical synaptic potentiation [100,116,117]. The long-term enhancement of PGO waves by the microinjection of the cholinergic agonist carbachol into the LDT/PPT may also lead to long-term changes in the regulatory systems of the brain [118]. Collectively, these findings underscore the important functions of PGO waves during REM sleep in the plasticity of the brain, in learning, and in the consolidation of memory during early childhood development and throughout life.

### 3.4. Mood Regulation

Because REM sleep is thought to play an important role in emotional memory processing, disrupted REM sleep may be an important contributor to the pathophysiology of emotion-based disorders such as major depression and PTSD [119]. Major depression is extremely common and is one of the leading causes of disability worldwide. Sleep disturbances are typical of most patients with major depression and are a core symptom of the disorder. Polysomnographic indices document objective changes in sleep continuity, slow-wave activity reduction, and REM sleep alterations, such as a shortening of REM sleep latency, prolongation of the first REM sleep period, and increased REMs density [95,120]. In addition to neurotransmitter imbalances, anatomical changes in brain structure and organization have recently been implicated as predisposing factors for major depression [121]. 

It has been proposed that PGO waves enhance synaptic plasticity in the areas that they pass through [70]. This includes the hippocampus and the amygdala [69,122]. Several studies have found that high amygdala reactivity is associated with an increased risk for the development of major depression [123]. The pons receives amygdala axonal projections [124], and the electrical stimulation of the amygdala increases the density of the PGO wave during REM sleep [125]. The antidepressant drugs (norepinephrine or serotonin reuptake inhibitors) reduce the density of the PGO waves [126]. The inhibition of the generation of PGO waves may have an antidepressant effect [126]. The hippocampus, which plays a central role in mood dysregulation and neurogenesis, appears to be associated with the behavioral symptoms of major depression. The negative effects of REM sleep disruption on hippocampus-dependent cognitive functions may be due to a decrease in adult hippocampal neurogenesis in humans [127]. Decreased functional connectivity at limbic cortical levels, particularly in the prefrontal, anterior cingulate, and insula, altered amygdala microstructure, and decreased claustrum volume have been reported following major depression [121,128,129,130]. 

Sleep complaints are virtually universal in PTSD patients. Although variations in the timing and/or amount of REM sleep have not been consistently observed in PTSD patients, other subtle differences in REM sleep are emerging, including an increased REMs density and REM sleep fragmentation [131,132]. Although restoring normal REM sleep may benefit extinction learning in PTSD [133], REM sleep may also have the opposite effect on the subsequent expression of learned fear during memory consolidation after trauma [119]. In other words, during the initial formation of traumatic memories, it is possible that REM sleep suppression could have a therapeutic effect on learned fear in the early stages of memory consolidation, while having a detrimental effect when applied at later stages of the disorder. The medial prefrontal inhibitory pathways that mediate fear circuitry in the amygdala likely play a key role in this effect [119]. 

The interconnectivity of amygdala and brainstem regions associated with REM sleep activity suggests reciprocal roles for emotion regulation and REM sleep onset, and waking exposure to emotional stimuli significantly affects subsequent REM sleep [119]. One study [134] found that fear conditioning enhanced PGO waves while suppressing REM sleep in rodents. It is likely that learning-dependent amygdala activity also influences REM sleep, as activity in the central amygdaloid nucleus enhances PGO waves. Presenting fear-related cues following fear conditioning also leads to significant increases in Fos expression in the amygdala and brainstem [135], while also leading to significant reductions in subsequent REM sleep. It has also been hypothesized that PTSD-related amygdala hyperresponsivity, such as PGO wave activity, may be a factor in PTSD-related sleep disturbances due to the reciprocal role of emotion regulation and REM [136]. Thus, abnormal REM sleep activity following trauma exposure may predict the subsequent development of PTSD. Enhanced REM sleep would bias memory consolidation toward an increased storage of negative content, suggesting a dysfunctional attenuation of emotional tone [95].

An interesting hypothesis that REM sleep constitutes a programming system that helps to maintain the process of psychological individuation was proposed by Jouvet [94]. As such, the PGO wave system can be compared to an internal programming system. Namely, during REM sleep, the brain is subjected to the endogenous programming of the PGO system. This programming would either reinforce or erase the synaptic circuitry related to emotional memory that was established during the previous waking state and can only be inferred by the theta rhythm that occurs during REM sleep. However, the role of PGO wave density during REM sleep in the development of mood disorders needs further exploration. We further speculate that processes related to this internal programming system during REM sleep may enhance memory after an aversive emotional experience in mood disorders. 

## 4. Conclusions

Numerous studies suggest that the function of PGO waves extends beyond sleep physiology alone, and their role in REM sleep and visual perception more broadly may be a promising avenue for further study, including dreams. The activation–synthesis and AIM theories can be seen as attempts to map animal models of REM sleep biology onto the phenomenology of human dreaming. Regardless, the current validity of biopsychological concepts in cats and other animal models for the human does not fully elucidate PGO’s role in these dream theories. Nevertheless, they are the most widely cited and published neurobiological theories of dreaming today. Recent theoretical models, such as that recently proposed by Gott et al. [38], in which PGO waves are expected to play a central role, could relate to the existence of a synthetic imagination marker. This would be experienced exclusively during dreaming, which would correlate with physiological brain mechanisms that effectively facilitate the brain’s ability to override its own retinal input. Next, PGO waves relevance may be further supported by reflecting on the evolution of the proposed mechanism of REM sleep regulation and related structural functions such as emotion, memory, and learning. For the nonpharmacological treatment of depression, stress, and PTSD, this has potentially far-reaching implications.

This review may assist the vastly growing interest in REM sleep, after the pioneering article entitled “Regularly occurring periods of eye motility and concomitant phenomena during sleep” by Aserisnky and Kleitman [2]. Indeed, the phenomenon of PGO waves in pre-clinical and clinical research is still a dream to be fulfilled. 

## Figures and Tables

**Figure 1 brainsci-13-01350-f001:**
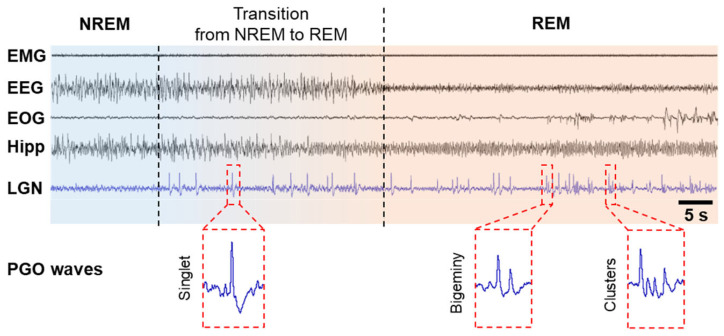
PGO waves in cats. PGO waves occur just before the onset of REM sleep, i.e., during the transition from NREM to REM, and during REM sleep period. PGO wave singlets occur predominantly during the transition from NREM to REM sleep and are not time-locked to rapid eye movements (REMs), but PGO wave clusters (≥3 waves) occur predominantly during REM sleep and correlate strongly with REMs and typical hippocampal theta oscillations. Abbreviations: EEG, electroencephalogram of the neocortex; EMG, electromyogram; EOG, electro-oculogram; Hipp, EEG of the hippocampus; LGN, EEG of the lateral geniculate nucleus.

**Figure 2 brainsci-13-01350-f002:**
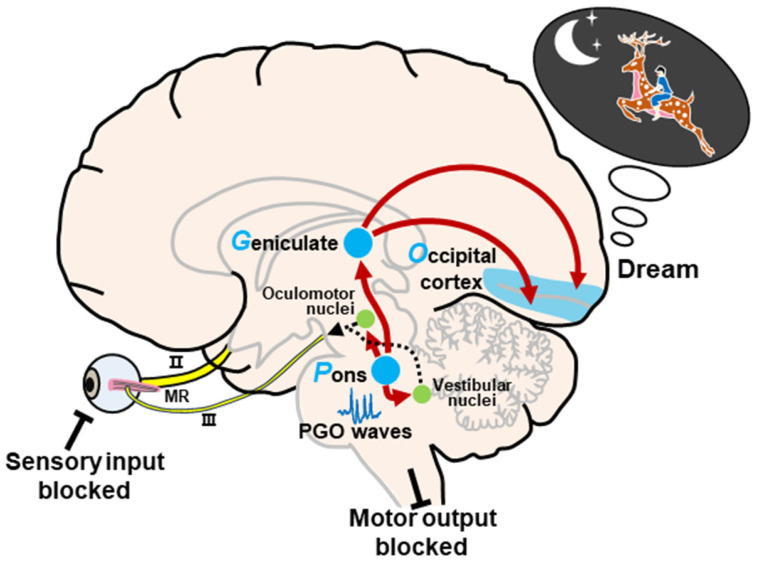
Schematic representation of the activation–synthesis model. Neural activity of PGO waves in the pons activates the lateral geniculate nucleus of the visual thalamus and the visual cortex. The passive synthesis of information (perceptual, conceptual, and emotional) generates dreams in the cortex. Meanwhile, the neural activity in the pons also activates the vestibular nucleus and oculomotor-related nuclei, such as the oculomotor nucleus, the trochlear nucleus, and the abducens nucleus, to induce REMs during REM sleep. In this state, external input and motor output are blocked. Abbreviations: II, optic nerve; III, oculomotor nerve; MR, medial rectus.

**Figure 3 brainsci-13-01350-f003:**
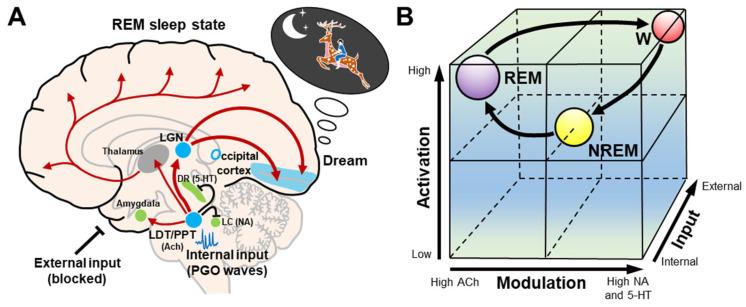
Schematic representation of the AIM model. (**A**) During REM sleep dreams, cholinergic neurons in the LDT/PPT are involved in the generation of internal input PGO waves, while serotonergic dorsal raphe nucleus and noradrenergic locus coeruleus neurons and external inputs are inhibited. These internal inputs activate the LGN and occipital cortex, and produce dreams. The LDT/PPT–thalamus–cerebral cortex pathway causes the desynchronization of the cerebral cortex. The LDT/PPT–amygdala pathway may be involved in mood regulation during dreams. (**B**) The cubic 3-dimensional model shows normal transitions within the 3-dimensional parameters (activation, input, and modulation) from wakefulness to NREM and then to REM sleep. Abbreviations: 5–HT, serotonergic; ACh, cholinergic; DR, dorsal raphe nucleus; LC, locus coeruleus; LDT/PPT, laterodorsal tegmental and pedunculopontine nuclei; LGN, lateral geniculate nucleus; NA, noradrenergic; NREM, non-rapid eye movement; REM, rapid eye movement; W, wakefulness.

## Data Availability

Not applicable.

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
