# Peer review of "The Ponto-Geniculo-Occipital (PGO) Waves in Dreaming: An Overview"

_brainsci, 2023, doi:10.3390/brainsci13091350_

Round 1

Reviewer 1 Report

This article is linked to an interesting and complex subject such as REM dreaming triggered by pons-geniculate-occipital waves (PGO) and a so-called scoping review that has steps to be followed, but apparently they were not (Mak S, Thomas A . Steps for Conducting a Scoping Review. J Grad Med Educ. 2022;14(5):565-567.). Instead, a narrative review seems to have been presented, but without well-defined objectives. Perhaps it would be interesting to start presenting the microstructure of REM sleep, mainly its phasic periods that are characterized by bursts of eye movements linked to PGO waves. Consequently, before evaluating the main subject, it is necessary to define what type of review was carried out. In any case, it was not clear that research results on PGO waves are based primarily on animal studies, particularly cats. However, human EEG studies have identified similar rapid eye movements and brain wave patterns associated with REM sleep, including PGO-like activity, and some studies have involved the implantation of electrodes in the brain, such as in individuals undergoing surgery to treat epileptic seizures. These electrodes can also be used to record brain activity during different stages of sleep, including REM sleep, providing insight into the generation of PGO waves and their role in dreaming. Furthermore, the use of functional imaging techniques such as fMRI and PET along with simultaneous EEG recordings can detect PGO waves, which allows researchers to correlate specific patterns of brain activity with the occurrence of PGO waves during REM sleep in humans.  However, much of the detailed understanding of PGO waves and their mechanisms still depends on animal studies. Therefore, findings from animal studies are often used as a basis for understanding human sleep physiology, but direct translation to humans may require further investigation and validation.

The automatic proofreader detected fewer problems in the English version, but the biggest problem is in the wording of the text, which does not seem to have clear objectives for this revision.

Author Response

Point 1

This article is linked to an interesting and complex subject such as REM dreaming triggered by pons-geniculate-occipital waves (PGO) and a so-called scoping review that has steps to be followed, but apparently they were not (Mak S, Thomas A . Steps for Conducting a Scoping Review. J Grad Med Educ. 2022;14(5):565-567.). Instead, a narrative review seems to have been presented, but without well-defined objectives. Perhaps it would be interesting to start presenting the microstructure of REM sleep, mainly its phasic periods that are characterized by bursts of eye movements linked to PGO waves. Consequently, before evaluating the main subject, it is necessary to define what type of review was carried out. In any case, it was not clear that research results on PGO waves are based primarily on animal studies, particularly cats. However, human EEG studies have identified similar rapid eye movements and brain wave patterns associated with REM sleep, including PGO-like activity, and some studies have involved the implantation of electrodes in the brain, such as in individuals undergoing surgery to treat epileptic seizures. These electrodes can also be used to record brain activity during different stages of sleep, including REM sleep, providing insight into the generation of PGO waves and their role in dreaming. Furthermore, the use of functional imaging techniques such as fMRI and PET along with simultaneous EEG recordings can detect PGO waves, which allows researchers to correlate specific patterns of brain activity with the occurrence of PGO waves during REM sleep in humans.  However, much of the detailed understanding of PGO waves and their mechanisms still depends on animal studies. Therefore, findings from animal studies are often used as a basis for understanding human sleep physiology, but direct translation to humans may require further investigation and validation.

Response 1: We thank Reviewer 1 for his/her pertinent comments, and we have made the following changes (all changed words and sentences throughout the manuscript are highlighted with red color) to all issues.

  • After our careful consideration, we think that “overview” which is a generic term – summary of the literature that attempts to survey the literature and describes its characteristics. Therefore, the title of the paper is changed to “The Ponto-Geniculo-Occipital (PGO) Waves in Dreaming: An Overview”.
  • We have added two paragraphs to the revised version to summarize the relevant research on human PGO in detail, as follows:

In humans, many invasive and non-invasive studies provide insight into how PGO waves occur during REM sleep and how they contribute to dreaming. For example, in two invasive studies in Parkinson's disease patients, deep brain stimulation electrodes were surgically implanted in the pedunculopontine nucleus of the pontine tegmentum [36] and the subthalamic nucleus (STN) [7]. The results showed that PGO-like waves were ob-served during REM sleep. Much like feline models, PGO wave singlets and clusters were recorded within STN during pre-REM and REM sleep [7]. A single-neuron study using depth-electrode EEG in patients with epilepsy found that potentials in the medial tem-poral lobe with a morphology similar to feline PGO waves were reliably observed time-locked to REMs, providing considerable support for the hypothesis that PGO waves propagate throughout the brain [42]. Several non-invasive technologies such as functional magnetic resonance imaging (fMRI) and positron emission tomography (PET) provide in-direct evidence to support the existence of PGO waves in humans. An fMRI study com-bined with polysomnographic (PSG) recording [43] showed that the pontine tegmentum, ventroposterior thalamus, primary visual cortex, putamen and limbic areas were activat-ed during REM sleep in association with REMs. Neural evidence for the existence of hu-man PGO waves and a link between REM and dreaming is provided by REM-related ac-tivation of the primary visual cortex in the absence of visual input from the retina. The ex-istence of PGO wave-like activity in humans was also supported by another study using PET combined with hemodynamic recordings of PSG activity [44]. This study found re-markable activity during REM in the ventroposterior thalamus and V1 of the occipital cortex, with additional activity in a number of limbic regions and the parahippocampal gyrus. This supports the hypothesis that PGO waves play an important role in the genera-tion of visual content during REM sleep.

Overall, much of the detailed understanding of PGO waves and their mechanisms still depends on animal studies. However, direct translation to humans may require fur-ther investigation and validation. Using noninvasive methods such as fMRI and magne-toencephalogram, the future of PGO wave research undoubtedly lies in broad studies that combine behavioral, pharmacological, and cognitive experiments in human subjects [38].

Point 2. The automatic proofreader detected fewer problems in the English version, but the biggest problem is in the wording of the text, which does not seem to have clear objectives for this revision.

Response 1: We asked Prof. Karen, a neuroscientist whose daily communication language is English, to make systematic linguistic revision of the full text. The revised manuscript has been carefully checked by authors again. We believe that the readability of our revised manuscript has been improved. 

Reviewer 2 Report

I have uploaded an edited version of the manuscript that contains my comments embedded in added sticky notes. 

most of the English is ok but there are a couple of runon and incomplete sentences. Also, it is not at all clear what Scoping means in the context of a "Scoping Review"  this might require some additional attention.

Author Response

We thank Reviewer 2 for his/her meticulous work. In response to these questions, we have conducted point-by-point answers (the red words in the revised manuscript are the mark of the changes) as follows:

Point 1. Line 30-32, This is a runon sentence -- make it two --  "...dreams. This..."

Response 1: Thank you for this comment. This sentence is divided into two sentences “Sigmund Freud’s “The Interpretation of Dreams” [1] initially developed the most prominent psychoanalytic theory of dreams. This model has been quite influential in sleep research and continues to have strong adherents to this day.” in the updated manuscript, line 31-33.

Point 2. Line 50, Why not cite Fernandez-Medoza, et al., 2009, Evidence of subthalamic PGO-like waves during REM sleep in humans: a deep brain polysomnographic study"

Response 2: Thank you for capturing this oversight. This paper has been cited in the revised manuscript as reference 7.

Point 3. Line 55, why is Ref. # 21 listed here. It seems unrelated to the topic of this sentence.

Response 3: Thank you for pointing out this error. We originally wanted to cite Hobson’s 2009 paper, so we renumbered the reference as 22.

Point 4. Line 56-57, Awkward sentence structure. Are only some PGO waves involved the proposed neural mechanisms of dreaming?

Response 4: Thank you for this comment. This sentence is changed to “This review summarizes the PGO waves involved in the proposed neural mechanisms of functioning such as dreaming and memory. The hypothesized physiological roles between PGO waves and performance are discussed.” in the updated manuscript, line 57-59.

Point 5. Line 60, why not just " PGO waves"

Response 5: We agree this suggestion and make change.

Point 6. Line 69, Why not also cite Fernandez-Mendoza, et al. 2009?

Response 6: Same as “point 2”, we have cited this article in the revised manuscript.

Point 7. Line 121, Still wondering why Ref. #21 is included in this discussion? in error??

Response 7: Thank you for capturing this oversight. We have been changed it.

Point 8. Line 136-137, has suggested what? incomplete sentence.

Response 8: We apologize for the lack of this information. This sentence is changed to “In addition, the involvement of γ-aminobutyric acid (GABA) and glutamate have also been suggested to participate in this process.” in the updated manuscript, line 165-167.

Point 9. Line 173, which quality of PGO wave activity? if a specific characteristic of the PGO wave is altered then be specific about that specific "quality" of the PGO wave. If by "quality" you are referring to the number or density of PGO wave then it is one of those characteristics of the wave being measured not a specific wave quality, per se.

Response 9: Thank you for pointing this out, we agree with you. There are lots of the characteristics belong to the PGO wave, such as density, amplitude, interval between waves and so on. Density is one of those characteristics. We have made modifications: “It appears that the density of PGO wave activity is directly related to memory processes. A number of studies have reported an increase in PGO wave density following fear memory training in rats, which predicted overnight memory consolidation [70-73].” in the updated manuscript, line 259-262.

Point 10. Line 177, see previous comment.

Response 10: The sentence is changed to “The success of fear extinction was recently shown to be predicted by PGO wave density during REM sleep [74].” in the updated manuscript, line 262-263.

Point 11. Line 203, imprint??? Not sure this is a useful term without definition.

Response 11: The original term is “enduring embossing” (Ramirez-Villegas et al., 2021), Therefore, “imprint” replaced by “enduring embossing”.

Point 12. Line 230-231, This statement is a conclusion of the authors' apparently and does not necessarily follow for the prior sentences. It would be better presented as the speculation it is rather than a conclusion as state.

Response 12: We agree and have modified this sentence to “Furthermore, PGO waves may determine which memories are retained and which are erased.” in the updated manuscript, line 316-317.

Point 13. Line 233, this statement should have a reference to support it.

Response 13: Thank you for your comment. We have added a reference [20] in the updated manuscript, line 319.

Point 14. Line 250, remains

Response 14: Thank you for capturing this oversight. “…remain to be answered” is changed to “…remains unanswered”.

Point 15. Line 234-252, There is no mention of PGO waves in this paragraph and there is more to REM sleep activity than simply PGO waves. A better argument needs to be made here to support the notion that the phenomena mentioned herein are related to PGO waves.

Response 15: We thank for the reviewer’s suggestion. The relevant arguments have been updated in the revised manuscript as follows (in the updated manuscript, line 320-346): “The unlearning theory is supported by a growing number of studies. For example, during REM sleep, the postsynaptic dendritic spines of layer V pyramidal cells in the mouse motor cortex are eliminated during development and motor learning. On the other hand, critical spines are strengthened and maintained [89]. The hypothalamic MCH neurons are known to be involved in the control of REM sleep and mood [90]. An anatomical and functional study in cats found that MCH neurons project to cholinergic pontine neurons [91]. When MCH was microinjected into the nucleus pontis oralis, there was a significant decrease in latency to REM sleep and a significant increase in the amount of REM sleep, accompanied by increased PGO wave activity and its duration. This suggests that the MCH system is involved in the regulation of REM sleep by modulating neuronal activity in cholinergic pontine neurons [91]. However, it has recently been shown that MCH neurons are also involved in the unlearning mechanism of REM sleep [92]. Activation or inhibition of MCH neurons by optogenetics and chemogenetics impaired or improved hippocampal-dependent memory, respectively. Activation of MCH nerve terminals in vitro reduced firing of hippocampal pyramidal neurons by increasing inhibitory inputs. These results strongly suggest that activation of MCH projections to the hippocampus during REM sleep actively contributes to forgetting [92]. The activity of hippocampal neurons during REM sleep has been suggested to play a key role in unlearning [93]. REM sleep serves to maintain or strengthen memories until they are transferred out of the hippocampus, whereupon they should be erased from this space-restricted short-term memory factory so that these synapses can be used to encode new associative memories [93]. PGO waves that are phase-locked to the theta oscillation of the hippocampus during REM sleep may be involved in this process [94]. However, the question of whether REM sleep plays a more critical role in the processing of emotional and procedural memories than other types of memories remains unanswered. One attractive hypothesis is that REM sleep is responsible for erasing negative emotional memories, and that this function is dysfunctional in depressed patients [95].”

Point 16. Line 256-257, PGO wave may play an important role in brain development, but it is not the case that PGO wave are the main or only stimulation that is important for brain development. The authors would do well to make this distinction more clearly.

Response 16: Thank you for pointing this out, we agree with you. Besides the PGO wave, the twitch is another important stimulator in early life. Blumberg et al. concluded that more myoclonic twitches of skeletal muscles occurring during early life REM sleep trigger sensory feedback and therefore contribute to the establishment of the sensorimotor system (Blumberg et al., 2013). The sentence is changed to “PGO waves during REM sleep in development might be an important central nervous system (CNS) stimulator during a period when wakefulness is limited in time and scope and stimulation opportunities are few [94,97,98].” in the updated manuscript, line 350-352.

Point 17. Line 291, The authors are not careful in this section (Brain Development and Plasticity) to make a strong case for involvement of PGO waves in brain development that is distinct from activation in general that occurs during REM sleep. They should reconsider this section anew.

Response 17: We thank for the reviewer’s suggestion and reconciled this section on PGO waves in brain development. We revised manuscript as follows: “REM sleep is known to be particularly abundant during early development. At birth, half or more of our sleep time is occupied by REM sleep, compared to < 20% of sleep time in adults [96]. PGO waves during REM sleep in development might be an important central nervous system (CNS) stimulator during a period when wakefulness is limited in time and scope and stimulation opportunities are few [94,97,98]. The ascending impulses emanating from the brainstem during REM sleep may be required to promote neuronal differentiation, maturation, and myelination in higher brain centers [17]. That is, REM sleep deprivation during early life of animals has been used to understand some functional mechanisms of PGO waves in brain development. Selective REM sleep deprivation further eliminates endogenous stimulation of PGO waves [17]. For example, when REM sleep is dramatically reduced in postnatal rats over the course of 2 weeks, these REM sleep-deprived rats have reduced brain size, hyperactivity, anxiety, attention and learning difficulties in adulthood [99,100]. In addition, selective REM sleep deprivation for 1 week during the critical postnatal developmental period in kittens significantly reduces the size of noradrenergic neurons in the locus coeruleus [101] and the number of parvalbumin immunoreactive neurons in the LGN [102]. REM sleep deprivation in monocularly occluded kittens reduces neuronal size in the monocular segment of the LGN [103] and ocular dominance plasticity by inactivating a kinase critical for this plasticity [104]. Elimination of phasic PGO waves in the LGN during REM sleep enhances plasticity effects on cell size in the LGN [105]. This suggests that REM sleep neuronal activities may be necessary for normal LGN development. The development of PGO waves is further thought to contribute to the maturation of the thalamocortical pathway in early life [98,106]. Thus, the abundance of REM sleep in early life and its subsequent decline to lower levels in adulthood strongly suggests that REM sleep is an integral part of the activity-dependent processes that enable normal physiological and structural brain development [100,102,106-108].

Several lines of evidence have shown that PGO waves in REM sleep is associated with the regulation of neural plasticity [109-111]. Brain plasticity allows the preservation of the ability to change, adapt, and learn in response to different environmental experiences and new demands. These processes occur with sleep cycles throughout life and begin in response to REM sleep in late fetal and early neonatal life [112]. PGO wave-associated cells are good candidates for generating or modulating plasticity in various brain structures [71], as they discharge high-frequency spike bursts during pre-REM and REM sleep [34]. Indeed, activation of PGO wave-generating cells by cholinergic agonism induces changes to the electrical properties of PGO wave activity [113], accompanied by prominent behavioral effects [74,114]. Furthermore, activation of the PGO wave generator and the occurrence of PGO waves themselves are tightly correlated with increased levels of cAMP response element binding (CREB) proteins, brain-derived nerve growth factor (BDNF), and activity-regulated cytoskeletal protein (ARC) in structures including the hippocampus and amygdala [72]. An investigation of the functions and underlying mechanisms of REM sleep during postnatal development in mice revealed that REM sleep plays a fundamental role in establishing stable connectivity of synaptic circuits during development. Almost all newly formed synapses cannot be maintained without REM sleep [89]. In addition, REM sleep plays an important role in a developmentally regulated form of in situ long-term potentiation (LPT) that coincides with the visual critical period [115]. During the visual critical period, PGO waves, especially the migratory waves of neural activity that can resemble evoked visual activity in sensory cortex [40], triggers a series of intracellular cascades and the synthesis of several plasticity-related proteins that promote cortical synaptic potentiation [100,116,117]. The long-term enhancement of PGO waves by microinjection of the cholinergic agonist carbachol into the LDT/PPT may also lead to long-term changes in the regulatory systems of the brain [118]. Collectively, these findings underscore the important functions of PGO waves during REM sleep in the plasticity of the brain, in learning, and in the consolidation of memory during early childhood development and throughout life.” in the updated manuscript, line 348-401.

Point 18. Line 306-309, something is missing in this sentence. Perhaps: "A reduction in functional connectivity ... has been observed following major depression"

Response 18: Thank you for pointing this out. This sentence is changed to “Decreased functional connectivity at limbic cortical levels, particularly in the prefrontal, anterior cingulate, and insula, altered amygdala microstructure, and decreased claustrum volume have been reported following major depression [121,128-130].” in the updated manuscript, line 425-428.

Point 19. Line 347, what is the distinction between PGO wave quality and quantity????

Response 19: Thank you for your comment. The sentence is changed to “However, the role of PGO wave density during REM sleep in the development of mood disorders needs further exploration.” in the updated manuscript, line 460-462.

Point 20. Line 350, this whole section (Mood Regulation) only mentions the PGO wave and its role in mood regulation in terms of needing further exploration. No argument is made to support a specific role for PGO waves in regulating mood. This section should be reconsidered.

Response 20: Thank you for your comment, The relevant arguments have been updated in the revised manuscript as follows (in the updated manuscript, line 403-464): “Because REM sleep is thought to play an important role in emotional memory processing, disrupted REM sleep may be an important contributor to the pathophysiology of emotion-based disorders such as major depression and PTSD [119]. Major depression is extremely common and is one of the leading causes of disability worldwide. Sleep disturbances are typical of most patients with major depression and are a core symptom of the disorder. Polysomnographic indices document objective changes in sleep continuity, slow-wave activity reduction and REM sleep alterations, such as a shortening of REM sleep latency, prolongation of the first of REM sleep period, and increased REMs density [95,120]. In addition to neurotransmitter imbalances, anatomical changes in brain structure and organization have recently been implicated as predisposing factors for major depression [121].

It has been proposed that PGO waves enhance synaptic plasticity in the areas they pass through [70]. This includes the hippocampus and the amygdala [69,122]. Several studies have found that high amygdala reactivity is associated with an increased risk for the development of major depression [123]. The pons receives amygdala axonal projections [124], and electrical stimulation of the amygdala increases the density of the PGO wave during REM sleep [125]. The antidepressant drugs (norepinephrine or serotonin reuptake inhibitors) reduce the density of the PGO waves [126]. The inhibition of the generation of PGO waves may have an antidepressant effect [126]. The hippocampus, which plays a central role in mood dysregulation and neurogenesis, appears to be associated with the behavioral symptoms of major depression. The negative effects of REM sleep disruption on hippocampus-dependent cognitive functions may be due to a decrease in adult hippocampal neurogenesis in humans [127]. Decreased functional connectivity at limbic cortical levels, particularly in the prefrontal, anterior cingulate, and insula, altered amygdala microstructure, and decreased claustrum volume have been reported following major depression [121,128-130].

Sleep complaints are virtually universal in PTSD patients. Although variations in the timing and/or amount of REM sleep have not been consistently observed in PTSD patients, other subtle differences in REM sleep are emerging, including increased REMs density and REM sleep fragmentation [131,132]. Although restoring normal REM sleep may benefit extinction learning in PTSD [133], REM sleep may also have the opposite effect on the subsequent expression of learned fear during memory consolidation after trauma [119]. In other words, during the initial formation of traumatic memories, it is possible that REM sleep suppression could have a therapeutic effect on learned fear in the early stages of memory consolidation, while having a detrimental effect when applied at later stages of the disorder. The medial prefrontal inhibitory pathways that mediate fear circuitry in the amygdala likely play a key role in this effect [119].

Interconnectivity of amygdala and brainstem regions associated with REM sleep activity suggests reciprocal roles for emotion regulation and REM sleep onset, and waking exposure to emotional stimuli significantly affects subsequent REM sleep [119]. One study [134] found that fear conditioning enhanced PGO waves while suppressing REM sleep in rodents. It is likely that learning-dependent amygdala activity also influences REM sleep, as activity in the central amygdaloid nucleus enhances PGO waves. Presenting fear-related cues following fear conditioning also leads to significant increases in Fos expression in the amygdala and brainstem [135], while also leading to significant reductions in subsequent REM sleep. It has also been hypothesized that PTSD-related amygdala hyperresponsivity, such as PGO wave activity, may be a factor in PTSD-related sleep disturbances due to the reciprocal role of emotion regulation and REM [136]. Thus, abnormal REM sleep activity following trauma exposure may predict the subsequent development of PTSD. Enhanced REM sleep would bias memory consolidation toward increased storage of negative content, suggesting a dysfunctional attenuation of emotional tone [95].

An interesting hypothesis that REM sleep constitutes a programming system that helps to maintain the process of psychological individuation was proposed by Jouvet [94]. As such, the PGO wave system can be compared to an internal programming system. Namely, during REM sleep, the brain is subjected to the endogenous programming of the PGO system. This programming would either reinforce or erase the synaptic circuitry related to emotional memory that was established during the previous waking state and can only be inferred by the theta rhythm that occurs during REM sleep. However, the role of PGO wave density during REM sleep in the development of mood disorders needs further exploration. We further speculate that processes related to this internal programming system during REM sleep may enhance memory after an aversive emotional experience in mood disorders.”

Point 21. Line 356-358, how do these concepts undermine the PGO role???

Response 21: we have revised the sentence to “Although the current validity of biopsychological concepts in cats and other animal models for the human does not fully elucidate PGO’s role in these dream theories.” in the updated manuscript, line 470-472.

Point 22. Line 368, It is not clear what a "scoping review" refers to.

Response 22: Thank you for pointing this out, we have revised the title of the paper and the wording here.

Point 23. Line 369, the title of the paper should be in quotes.

Response 23: Thank you for your comment. We have put quotation marks in this sentence.

Round 2

Reviewer 2 Report

The manuscript is much improved as the authors have addressed each of the concerns of this and another reviewer.

  However, this reviewer has still identified a number of problematic sentences throughout the revised manuscript that would benefit from additional attention/rewriting. Some of this may have to do with English usage as opposed to outright conceptual problems. 

  This reviewer is attaching a PDF file of the revision with comments and suggested changes embedded therein.

  This reviewer has still identified a number of problematic sentences throughout the revised manuscript that would benefit from additional attention/rewriting. Some of this may have to do with English usage as opposed to outright conceptual problems. 

Author Response

We thank Reviewer 2 for his/her meticulous work. In response to these questions, we have conducted point-by-point answers (the red words in the revised manuscript are the mark of the changes) as follows:

Point 1. Line 13, not at all clear what "It" refers to. PGO waves are generated(the message?) in the pons (P) and can be recorded bilaterally in the LGN (L), occipital cortex (O), and other areas of the brain. See your lines 71 - 73 that expresses this more clearly.

Response 1: Thank you for this comment. This sentence is changed to “PGO waves look just like a message sent from the pons to the lateral geniculate nucleus of the visual thalamus, the occipital cortex, and other areas of the brain.” in the updated manuscript, line 13-15.

Point 2. Line 20, Only one theory is described in this sentence.

Response 2: Thank you for pointing out this potentially confusing sentence. This sentence is changed to “In this review, we summarize the role of PGO waves in potential neural circuits of two major theories, i.e. (1) Dreams are generated by the activation of neural activity in the brainstem; (2) PGO waves signaling to the cortex.” in the updated manuscript, line 18-20.

Point 3. Line 23, make this "their possible functions"

Response 3: It has been revised in the manuscript.

Point 4. Line 35-37, as written, the highlighted sentence does not make sense. It is not clear what distinction the author is making between "The concept that dreams arise from complex psychological mechanisms.." and how that is different from those underlying  "...the thoughts, feelings and memories of our typical waking experience".

Response 4: This sentence has already been deleted.

Point 5. Line 76, why is it "Therefore," the preceding sentence(s) do not necessarily lead into this sentence.

Response 5: We agree this suggestion and make change.

Point 6. Line 117, what about physiological measures????

Response 6: Thanks for your suggestion, “physiological” is inserted in this sentence.

Point 7. Line 168, PGO burst cells are not released. They may release cholinergic neurotransmitter (AcH) but the cells themselves are not released.

Response 7: Thank you for capturing this oversight. This sentence is changed to “This model postulates that the transition to the cholinergic state in REM sleep, during which cholinergic PGO burst cells in the brainstem release acetylcholine, alters the mnemonic capacity of the brain-mind and reduces the reliability of cortical circuits, increasing the likelihood that bizarre temporal sequences and associations are accepted as reality during dreaming [47].” in the updated manuscript, line 167-171.

Point 8. Line 474-475, If Gott et al. recently proposed such a model then it can't be a future model.

the sentence needs to be reworked accordingly.

Response 8: Thank you for pointing out the logical error. This sentence is changed to “Recent theoretical models, such as recently proposed by Gott et al. [38]” in the updated manuscript, line 473-474.

Point 9. Line 478-480, tThis sentence make little sense. The brain does not simultaneously balance inputs from the retina and the structures producing PGO-waves because they are mutually exclusive in time.

Response 9: Thank you for pointing this out, we agree with you. This sentence has already been deleted.

Point 10. Line 480, what is this "its" referring to? It is not clear from the context of this sentence.

Response 10: “its” replaced by “PGO waves” in the updated manuscript, line 478.
